# Spatial pattern and associated factors of timely vaccination in Ethiopia using EDHS-2016 data: A multilevel and spatial analysis

**Muluken Chanie Agimas** *, **Meron Asmamaw**, **Nebiyu Mekonen, Fantu Mamo, Daniel Alayu Shewaye**

College of Medicine and Health Science, Institute of Public Health, Department of Epidemiology and Biostatistics, University of Gondar, Gondar, Ethiopia

* mulukensrc12@gmail.com

## Abstract

### Background

Age-appropriate vaccination or vaccine timeliness is the administering of vaccines on the specified schedule of immunization. One of the qualities of the immunization program is an age-appropriate vaccine, it has become an ignored indicator of program performance. Even though age-appropriate vaccination is critical for child health, there are no national-level studies to generate conclusive and tangible evidence about the determination of timely vaccination in Ethiopia.

### Objective

To assess the spatial pattern and associated factors of timely vaccination in Ethiopia using EDHS-2016 data: A multilevel and spatial analysis.

### Method

Community based cross-sectional study design was employed from 18 January to 27 June 2016. To select the participants, two-stage cluster sampling was employedin the Ethiopian Demographic Health Survey 2016 data. Permission was obtained via online request by explaining the aim of this particular study from DHS international. A statistical package for social science-21 software was used for data cleaning, recoding, and analysis. Arc GIS 10.3 software was used to show the spatial variation of age-appropriate vaccination practices. A generalized linear mixed-effect model was used. For all models, intra-class correlation, a proportional change in variance, the log-likelihood test, and the Akaike information criterion were calculated. The best model was selected by the lowest value Akaike information criterion. Variables with a p-value less than 0.05 and a 95% confidence level were considered for the statistically significant association.

**Data Availability Statement:** Data cannot be shared publicly because of the data owner/DHS international restrict to share for third party. they

told us kept confidentiality. rather the data is available online. any one who want to access the data can access the data at https://dhsprogram.com/data/dataset_admin/index.cfm.

**Funding:** No specific funding for this work.

**Competing interests:** The authors have declared that no competing interests exist.

**Abbreviations:** AIC, Akaike Information Criteria; ANC, Antinatal Care; AOR, Adgusted Odds Ratio; BCG, Bacilli Calmati Gurrien; CI, Confidence Interval; DHS, Demographic Health Service; EDHS, Ethiopian Demographic Health Service; ICC, Intra-Class Correlation; OPV, Oral Polio Vaccine; PCV, Pnemococal Conjugated Vaccine; SNNPR, South Nation and Nationality of Peoples Region; TT, Tetanus Toxoide; VPDs, Vaccine Preventable Disease; WHO, World Health Organization.

## Result

The spatial distribution of age-appropriate vaccination practice in Ethiopia was non-randomly distributed with the global Moran's I value of 0.22 (p-value <0.001). The prevalence of age-appropriate vaccination practice in Ethiopia was 33.8%. Women who receive $\geq 2$ TT vaccines (AOR = 1.48; 1.22, 1.79), women who lived in rural residences (AOR = 0.77; 0.62, 0.96), gave birth at a health facility (AOR = 1.2; 1.12, 1.37), ANC follow up (AOR = 174; 1.45, 2.1), richest (AOR = 1.65; 1.15, 2.37), richer (AOR = 1.72; 1.3, 2.22), middle-level income (AOR = 1.65; 1.16, 2.36), poorer (AOR = 1.47; 1.11, 1.96) were the factors of age-appropriate vaccination practice.

## Conclusion

The spatial distribution of age-appropriate vaccination practice in Ethiopia was non-randomly distributed across the regions. Age-appropriate vaccination practice was low in Ethiopia. Wealth index, usual caretaker of the child, ANC utilization, history of TT vaccination, place of delivery, and residence were associated with age-appropriate vaccination practice.

## Introduction

Age-appropriate vaccination or vaccine timeliness is the administering of vaccines within the specified schedule of immunization and the world health organization recommends that children should better get vaccines in the appropriate time frame of the vaccination in their first year of life [1]. Children can protect themselves against the disease adequately when they take the vaccine at an appropriate period of time. The coverage of the vaccine is effective in disease protection when the vaccine dose is administered in timely vaccines [2]. Around the globe, starting from 1980 timely vaccination coverage has significantly progressed over time. Consequently, 2 to 3 million vaccine-preventable deaths have been prevented every year, but still, 1.5 million children die each year by easily vaccine-preventable diseases [3]. There is a need to expand the age-appropriate vaccination coverage to achieve the full benefits of vaccinations because unappropriated age vaccines can be the basis of improper disease protection [4, 5]. Because of this reason, the world health organization (WHO) recommends that the vaccine should take within 1 year schedule to maximize the protective effect of the vaccine [6]. In contrast to the aforesaid, administering the vaccine before the appropriate age can fail to produce specific antibodies against the antigen due to a sub-optimal seroconversion rate [7]. Consistently, in low- and middle-income countries, the problem of supply chain management, health service, and the performance of the health care provider is the challenge of age-appropriate vaccine practice [8].

In sub-Saharan African countries, including Ethiopia, vaccination delays have been a bottleneck issue [9]. Even though factors of vaccine timeliness vary with the study context [10], pieces of evidence from the previous study revealed that the risk factors of vaccine timeliness are home delivery [11], low-education attainment and below four antenatal care visits [12], unplanned pregnancy and child male sex [13], highest mothers/caregivers age [14], vaccine hesitancy [15], being a multiparous mother [16], and rural children and poorest quintile [8]. Like other countries, in Ethiopia, age-appropriate vaccination is the key strategic activity to reduce easily preventable childhood mortality from measles, pneumonia, diarrheal diseases, and other vaccine preventable diseases [17]. But still childhood mortality due to vaccine preventable diseases (VPDs) is high and as 2019 WHO report revealed, among the ten countries

Ethiopia ranked fifth with the highest defenseless childhood immunization [18]. Studies so far in Ethiopia have focused solely on full vaccination coverage. Studies on timelines of vaccination are few. Among those few studies reported about child vaccine timeliness before 7 years ago are; in a study done in the Menz Lalo district of Northeast Ethiopia the age appropriate vaccination practice was 6.2% [13] and 23.9% in the Toke Kutaye district, central Ethiopia [19]. Even though age-appropriate vaccination is critical for child health, there are no national-level studies to generate conclusive and tangible evidence about the determinants of timely vaccination in Ethiopia for policymakers, planners, non-governmental organizations, health facilities, and educators. Thus, the aim of this national-level study was to assessthe spatial variationand determinants of age-appropriate vaccination practice in Ethiopia.

## Methods

### Specific objectives

To determine the prevalence of age appropriate vaccination practice in Ethiopia using EDHS-2016 data

To determine the spatial variation of age appropriate vaccination practice in Ethiopia using EDHS-2016 data

To identify factors associated with age appropriate vaccination practice in Ethiopia using EDHS-2016 data

### Data source, study setting, study design, study population and sampling technique

Community based cross-sectional study design was employed from 18 January to 27 June 2016. Ethiopia is a sub-Saharan African country. According to 26 February, 2023 world meter reports, the total population of Ethiopia is 123,001,400. Of the total population, around 21.3% of the population is live in urban areas [20]. In Ethiopia, there are nine regions and two city administrations. All regions are divided into two geographical areas, urban and rural. The study population was all mothers who had a live birth in Ethiopia 12–35 months preceding the survey, those mothers (15–49 years) in the selected enumeration areas were the source population [21]. The numbers of women included in the survey were 15,683. Among these women finally a total of 4083 women having a child with aged 12–35 month were interviewed about age appropriate vaccination practice from January 18 to June 27. All women having a child with age 12–35 months and able to read write or/and hear were included in the study. To select the participants, two-stage cluster sampling was employed. The detailed sampling procedure was recorded in EDHS 2016 data. Permission was obtained via online request by explaining about the aim of this particular study from the DHS international, and the data set was downloaded from www.measuredhs.com.

### Study variables

**Age appropriate vaccination practice.** Is considered to be age appropriate vaccination practice/timely vaccinated if the child received BCG within the first 4 weeks, OP1, Penta 1, PCV1, and Rota 1 from 6 weeks to 10 weeks, OPV 2, Penta 2, PCV 2, and Rota 2 from 14 weeks to18 weeks, measles vaccination at 9 month and labeled as yes (code = 1) On the contrary, the child was considered as early vaccinated when the child received at least one dose of the vaccine below the minimum recommended age for each antigen and considered as delayed vaccination when the child received at least one dose of vaccine above the maximum recommended age and labeled as no (code = 0) [22, 23]. Residence, wealth index, age, educational

level, religion, ethnicity, sex of the child, marital status, place of delivery, ANC follow-up, numbers of a living children, birth order, receive TT dose, region, own mobile phone and current pregnancy wanted were the independent variables.

## Data processing and analysis

Data was extracted from 2016 EDHS data. Data wasextracted and analyzed from November-1 to December 30/2022. A statistical package for social science (SPSS)-21 software was used for data cleaning, recoding, and analysis. Age-appropriate vaccine practice across clusters was checked for variety. For complex surveys and unequal probabilities of selection, sampling weight was applied to all analysis procedures. Because of the national level of data, a hierarchical and clustered multilevel analysis model was used. Svy" command was used as the sampling weight of cluster sampling.

Intra-class coefficient (ICC) was used for the measurement of variation across regions with value of 18.7%. Analysis was undergone in four models; null model (model of without independent variables), model-I (individual level/level I predictors), model-II (group level/level II predictors), model-III (both model-I and model-II/mixed effect model). For all models ICC, a proportional change in variance (PCV), log-likelihood test, and Akaike information criterion (AIC) model were calculated (Table 1). The best model was selected by AIC or using the lowest value of AIC. A generalized linear mixed-effect model was used. Variables with a p-value less than 0.05 and a 95% confidence level were considered for statistical significance.

## Spatial autocorrelation

To analyze Moran's I, ArcGIS version 10.3 was used. The global Moran's I works on the value of 1 to -1. The value closes to −1 shows dispersed age appropriate vaccination practice. Whereas Moran's I value closest to +1 shows clustered age appropriate vaccination practice and the value is 0 shows randomly distributed age-appropriate vaccination practice. Moran's I (P-value < 0.05) was used to declare the statistical significance of spatial autocorrelation.

## Hot spot analysis (Getis-Ord Gi* statistic)

To determine how spatial autocorrelation differs across the areas of Ethiopia, Getis-Ord Gi* statistics were calculated. To evaluate the significance of clustering Z-score was calculated and the p-value was computed to evaluate the significance of clustering [28].

## Spatial interpolation

Knowing about a certain health condition in all parts of the country is impossible because of time and resource constraints. So the method of interpolation is very important to predict unsampled areas based on sampled data. To do this a technique of ordinary kriging spatial interpolation was used to predict age-appropriate vaccination practice in the un-sampled areas

**Table 1. Multilevel fixed effect model of individual and community level factors predicting age appropriate vaccine coverage in Ethiopia EDHS 2016, Ethiopia (n = 4083).**

| Random effect | Null model | Model I | Model II | Model III |
|---|---|---|---|---|
| Variance | 0.75 | 0.39 | 0.51 | 0.38 |
| ICC | 18.7% | 10.8% | 13.5% | 10.4% |
| PCV (%) | Reference | 48% | 32% | 49.3% |
| Log likelihood | -2955.9 | -2566.5 | -2888.8 | -2563.66 |
| AIC | 5915 | 5146 | 5785 | 5145 |

in the country based on sampled enumeration areas of Ethiopia. Things that are close together tend to be more related than things that are far apart. Therefore, the spatial dependence of the age appropriate vaccination practice nearby locations was considered. Without the degree of spatial dependence, it is impossible to represent continuous geographic phenomena in digital form.

### SaT scan analysis

Bernoulli-based model spatial SaT scan statistics using Kuldorff Sat Scan version 9.6 software was used to evaluate the statistical significance cluster of age appropriate vaccination in Ethiopia. The default maximum spatial cluster size of 50% was used.

### Ethical consideration

Since it is secondary data from EDHS, ethical clearance was not applicable but permission for further analysis was obtained from the DHS international. But informed written consent was obtained during EDHS survey to assure their willingness of participation in the 2016 national survey.

## Results

### Socio-demographic characteristics

A total of 4083 women having aged 12–35 month child in preceding the survey were included and interviewed about age appropriate vaccination practice. About 2313 (61.6%) were within the age group of 25–34 years, and 2539 (62.3%) and 1081(26.5%) were with no education and primary education respectively. Furthermore, the majority of the participants 3299 (80.8%) were from rural residence (Table 2).

### Obstetrics and other health service utilization related factors

In this study 2562(62.7%) women gave birth at home and 3000 (73.5%), 553(13.5%) of them were not pregnant and don't know about their pregnancy status respectively during the survey. Furthermore, 1285 (31.4%) of women were have four and above child (Table 3).

### Overall age appropriate vaccination practice in Ethiopia

In the current study, age appropriate vaccination practice in Ethiopia was 1383 (33.8%) [95% CI; 32.3, 35.3] (Fig 1).

### Age-appropriate vaccine for each vaccine

According to EDHS-2016 data the highest numbers of child (4382) take polio-2 at appropriate age (Fig 2).

### Spatial distribution of age appropriate vaccination practice in Ethiopia

A total of 620 clusters were used for spatial analysis of age appropriate vaccination practice in Ethiopia. The red color indicates a zero proportion of age appropriate vaccination practice whereas the yellow color indicates a range of 0.001% to 100% age appropriate vaccination practice areas (Fig 3).

**Table 2. Socio-demographic characteristics of women having 12–35 aged children in Ethiopia using EDHS-2016 (n = 4083).**

| Variables | Category | Frequency | Percent (%) |
|---|---|---|---|
| Sex of child | Male | 2082 | 51% |
| | Female | 2001 | 49% |
| Women age | 15–24 | 1105 | 27.1% |
| | 25–34 | 2313 | 61.6% |
| | > = 35 | 665 | 16.3% |
| Marital Status | Single | 31 | 0.8% |
| | Married | 3802 | 93.1% |
| | Living with partner | 37 | 0.91% |
| | Divorce | 126 | 3.1% |
| | Widowed | 50 | 1.2% |
| | No education | 2539 | 62.3% |
| Education status | Primary school | 1081 | 26.5% |
| | Secondary school | 300 | 7.3% |
| | Vocational/technique | 99 | 2.4% |
| | Higher education | 64 | 1.5% |
| Religion | Orthodox | 1243 | 30.4% |
| | Muslim | 2040 | 50% |
| | Protestant | 698 | 17.1% |
| | Traditional | 42 | 1% |
| | Others* | 31 | 0.7% |
| | Diredawa | 212 | 5.2% |
| Region | Tigray | 420 | 10.3% |
| | Afar | 416 | 10.2% |
| | Amhara | 365 | 9% |
| | Oromia | 604 | 14.7% |
| | Somali | 549 | 13.4% |
| | Benishangul | 329 | 8.1% |
| | SNNPR | 485 | 11.9% |
| Residence | Urban | 784 | 19.2% |
| | Rural | 3299 | 80.8% |
| current pregnancy | Yes | 530 | 13% |
| | No | 3000 | 73.5% |
| | Don't know | 553 | 13.5% |
| Use mobile | Yes | 995 | 24.4% |
| | No | 3088 | 75.6% |
| | Poorest | 1536 | 37.6% |
| Wealth index | Poorer | 676 | 16.6% |
| | Middle | 552 | 13.5% |
| | Richer | 494 | 12.1% |
| | Richest | 825 | 20.2% |

**Note**: others* = catholic

## Spatial autocorrelation

The spatial autocorrelation analysis revealed that the spatial distribution of age appropriate vaccination practice in Ethiopia was none randomly distributed. The global Moran's I value 0.22 (p-value <0.0001) which infers that there was significantly clustering of age appropriate vaccination practice in Ethiopia across the region (Fig 4).

**Table 3. Obstetrics/birth and other health service utilization related factors of women having 12–35 aged children in Ethiopia using EDHS-2016 (n = 4083).**

| Variables | category | Frequency | Percent (%) |
|---|---|---|---|
| Birth order | 1 | 894 | 22% |
| | 2–4 | 1746 | 42.8% |
| | > = 5 | 1443 | 35.2 |
| current pregnancy | Yes | 530 | 13% |
| | No | 3000 | 73.5% |
| | Don't know | 553 | 13.5% |
| Is current pregnancy wanted | Yes | 406 | 76.6% |
| | No | 124 | 23.4% |
| Numbers of live birth | 1–2 | 789 | 19.3% |
| | 3–4 | 1974 | 48.3% |
| | >4 | 1285 | 31.4% |
| | Yes | 2193 | 65.2% |
| ANC utilization TT dose | No | 1171 | 34.8% |
| | Not received | 2228 | 54.6% |
| | 1 | 371 | 9.1% |
| | > = 2 | 1484 | 36.3% |
| Usual care taker of the child place of delivery | Mother | 3790 | 92.8% |
| | Other than mother | 293 | 7.2% |
| | Home | 2562 | 62.7% |
| | Health facility | 1521 | 37.3% |

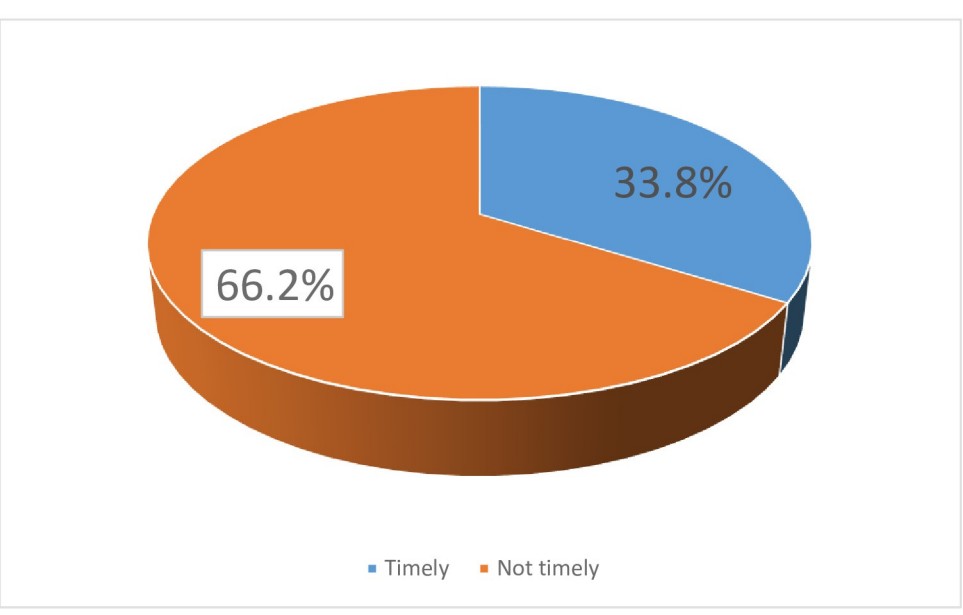

**Fig 1. Age appropriate vaccination practice among Ethiopian women based on EDHS-2016 data (n = 4083).**

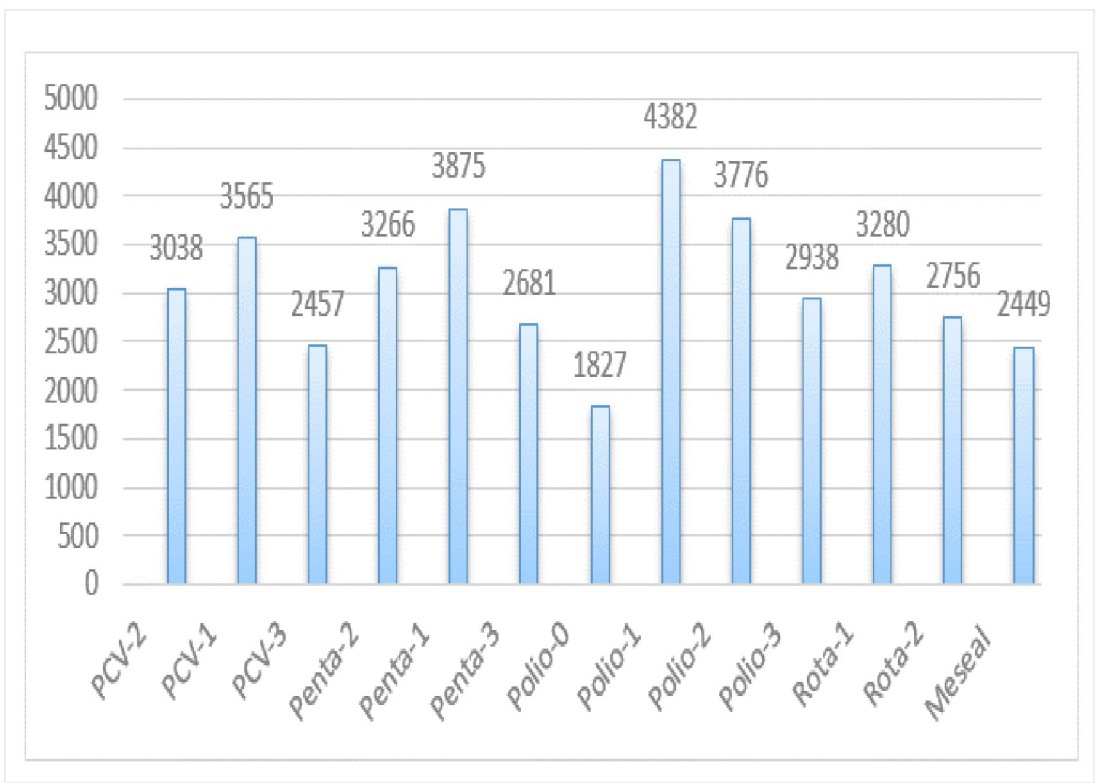

**Fig 2. Age appropriate vaccination practice for each vaccine among Ethiopian women based on EDHS-2016 data (n = 4083).**

### Hot and cold spot analysis of age appropriate vaccination practice in Ethiopia

In the Getis-OrdGi* statistical analysis, significant hotspot areas (high probability of age appropriate vaccination practice) were clustered in Addis Ababa, Northern part of Tigray, south western part of Benishangul Gumuz. While the cold spot areas (low-probability of age appropriate vaccination practice) were found in Northern part of Somali, eastern part of Amhara, North eastern part of SNNPR, western part of Oromia and western part of Gambela (Fig 5).

### Spatial interpolation

The interpolation analysis (spatial ordinary kriging) predicted highly practiced regions for age appropriate vaccination predication of high-practice areas was indicated by red predictions. Addis Ababawas predicted as more practice the age appropriate vaccinationas compared to other regions. In contrast, North West Gambela, Western part of Oromia, northern, southern and south eastern Somali, northern and south eastern part of Afar, northern and eastern part of Amhara were predicated as having the least practice of age appropriate vaccination as compared to other regions (Fig 6).

### Sat scan analysis

As the Fig 7 shows, the red window, the green window, the yellowand the blue windows on the map were the significant clusters of age appropriate vaccinations practice in Ethiopia. The

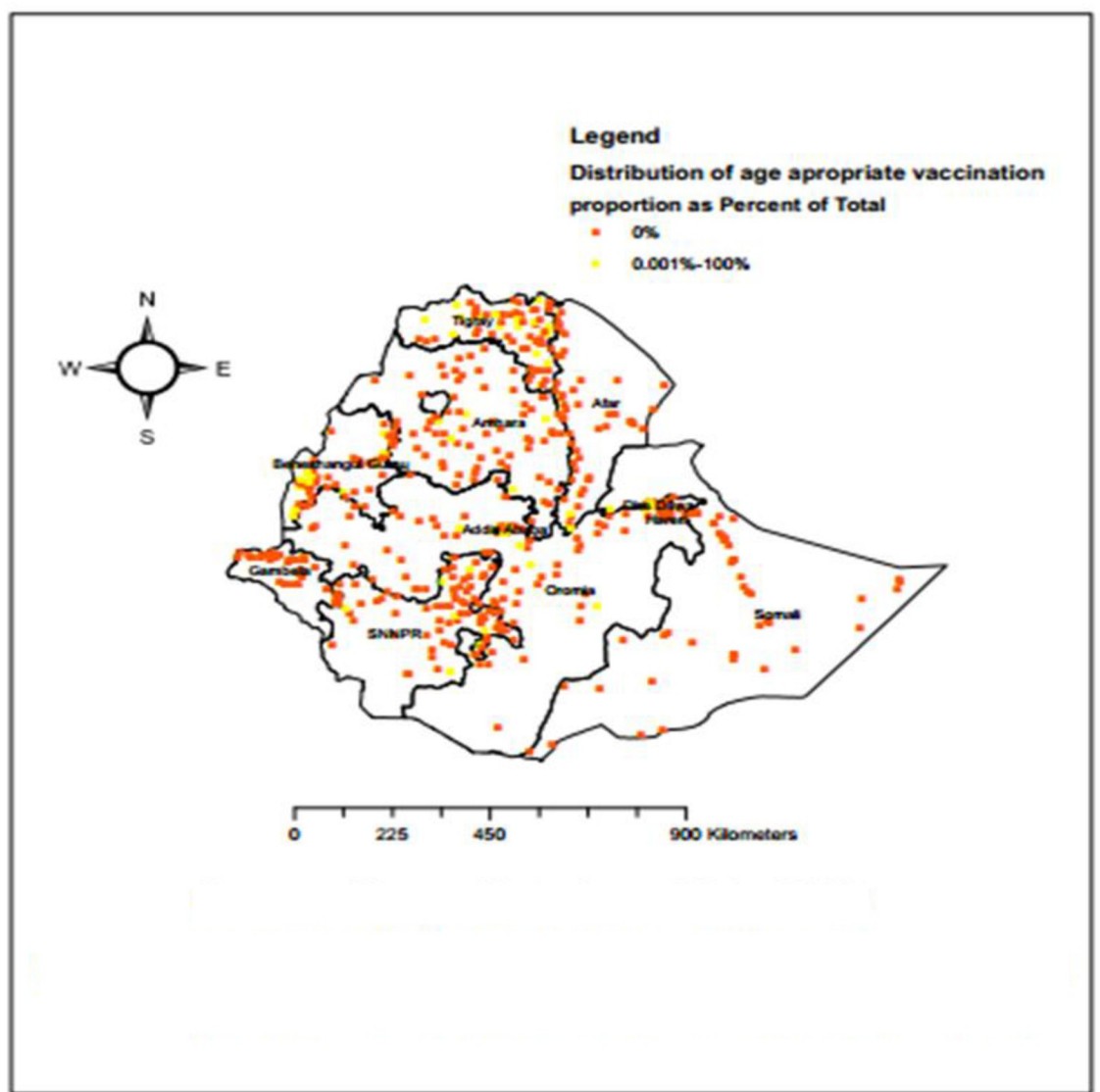

**Fig 3. Spatial distribution of age appropriate vaccination practice in Ethiopia using EDHS -2016 (n = 4083).** Source: Shape file is from CSA, 2013.

total number of clusters found statistically significant was 63. Of the total statistically significant clusters, 29 of them were primary (most likely) cluster types; the rest 34 clusters were secondary, tertiary, and quarterly clusters. The primary cluster was located at 9.519402 N, 35.863468 E within a 131.73 km radius. The age appropriate vaccination practice in the most likely clustered area was 5.4 times higher than outside the window (RR = 5.4, LLR = 20.5, P-value of 0.0000012) (Table 4, Fig 7).

## Determinants of age appropriate vaccination practice

Variables associated with age-appropriate vaccination practice were selected by first using the Bivariable mixed-effect logistic regression model. Variables such as wealth index, the number of TT doses received, and ANC utilization were the candidate variables for the multivariable

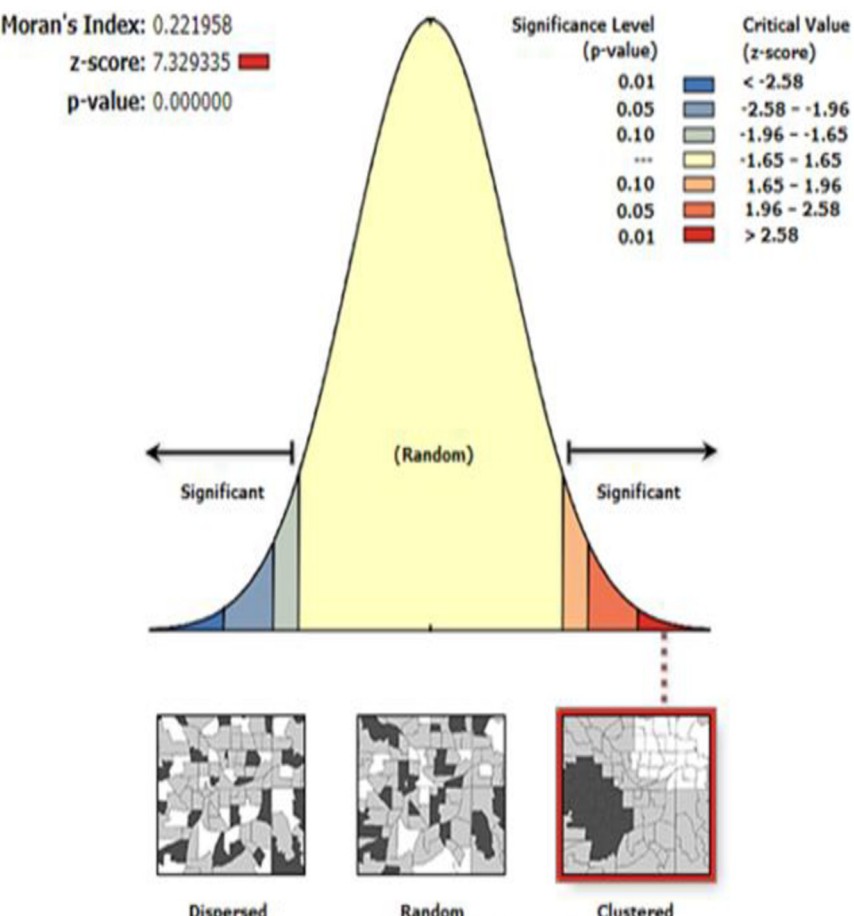

**Fig 4. Spatial autocorrelation of age appropriate vaccination practice in Ethiopia using EDHS 2016 data (n = 4083).**

mixed-effect logistic regression model with a p-value of 0.2 (model-I/level one predictor). On the other hand, in level two predictors (model II) variables such as residence of the women and place of delivery were significantly associated with age-appropriate vaccination practice.

Finally based on AIC, model III (mixed effect model) was the best model to identify predictors of age-appropriate vaccination practice since AIC and deviance were the lowest in model III. As such, the odds of age-appropriate vaccination practice among women who took two and more doses of the TT vaccine were 1.48 (AOR = 1.48; 1.22, 1.79) times more likely than those who did not take the TT vaccine. The odds of age-appropriate vaccination practice among women who lived in rural residences were reduced by 23% (AOR = 0.77; 0.62, 0.96) as compared with women living in urban. Similarly, women who gave birth at a health facility were about 1.2 (AOR = 1.2; 1.12, 1.37) times more likely to practice age-appropriate vaccines than those who gave birth at home. The odds of age-appropriate vaccine practice among women having ANC follow-up were 1.74 (AOR = 174; 1.45, 2.1) times more likely to practice age-appropriate vaccination than their counterparts. Furthermore, richest women, richer women, middle-level income women, and poorer women were 1.65 times (AOR = 1.65; 1.15, 2.37), 1.72 times (AOR = 1.72; 1.3, 2.22), 1.65 times (AOR = 1.65; 1.16, 2.36), 1.47 times (AOR = 1.47; 1.11, 1.96) more likely to practice age appropriate vaccination than the poorest women respectively (Table 5).

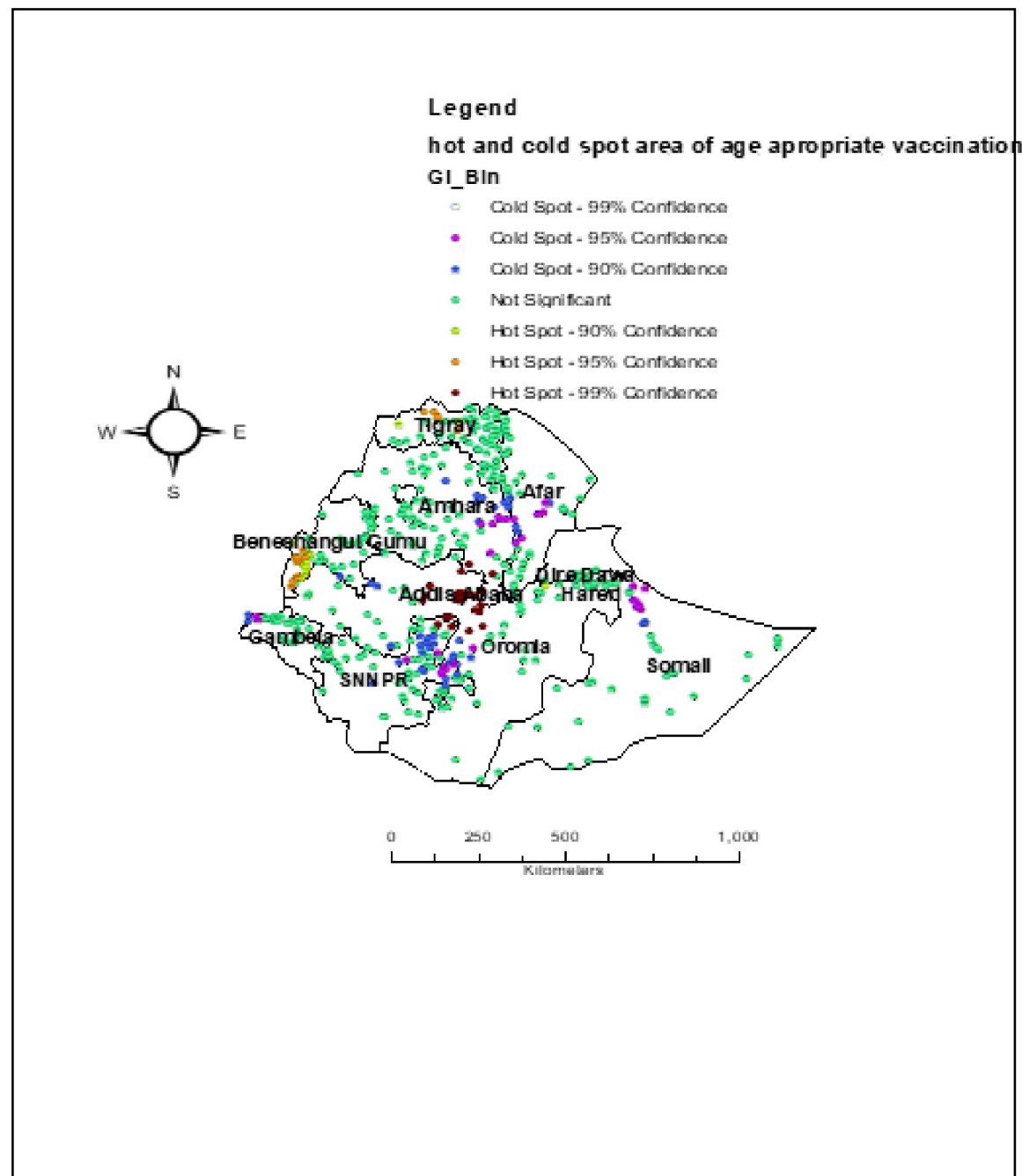

**Fig 5. Hot and cold spot area of age appropriate vaccination practice in Ethiopia using EDHS 2016 (n = 4083).** Source: Shape file is from CSA, 2013.

## Discussion

Age-appropriate vaccination practice is a vital strategy to prevent VPDs. In this study, an attempt has been made to assess the spatial variation and determinates of age-appropriate vaccine practice among Ethiopian women who have a child aged from 12–35 months. The spatial variation of age appropriate vaccination practice in Ethiopia is non-randomly distributed with

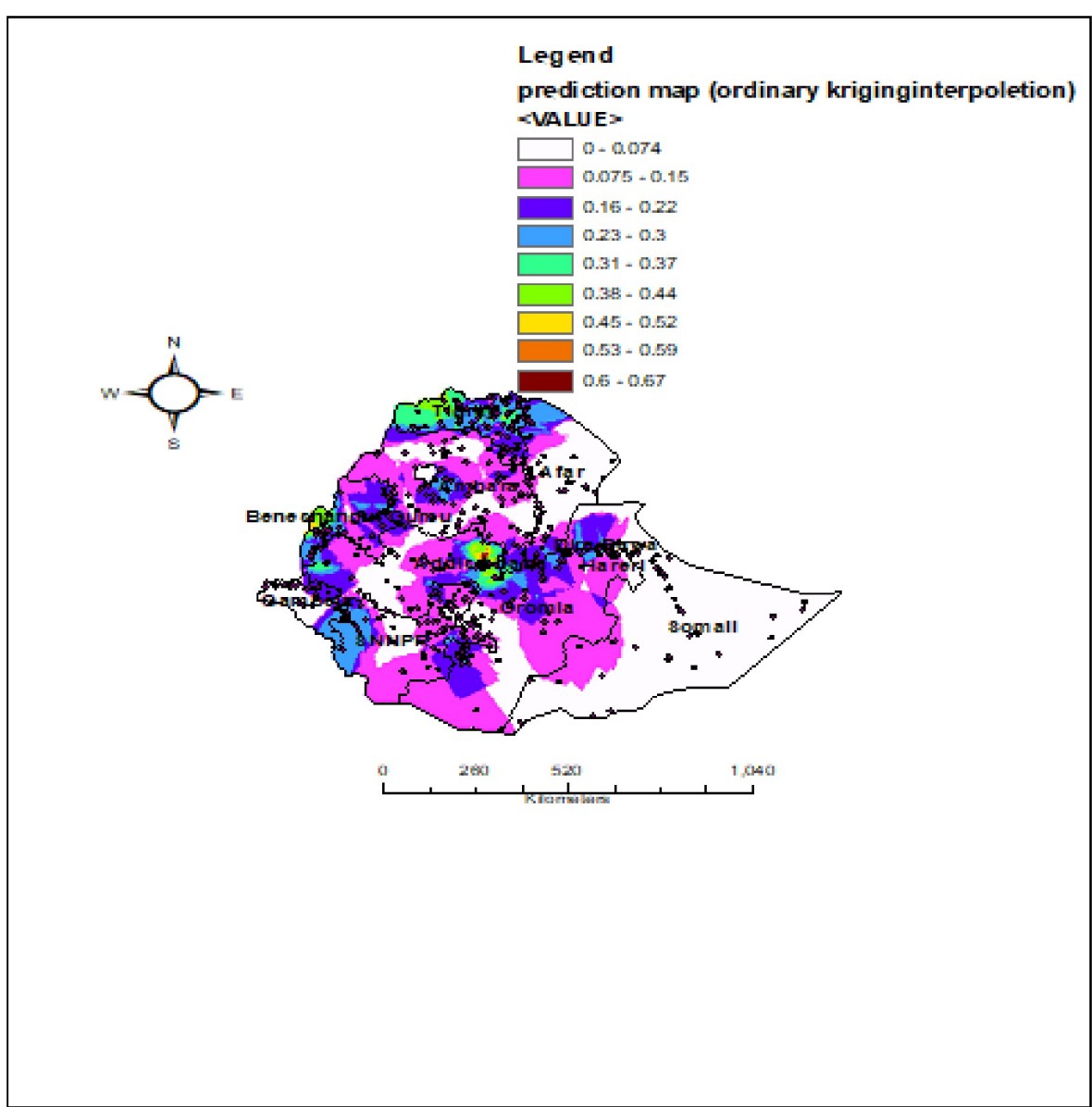

**Fig 6. Spatial interpolation of age appropriate vaccination practice in Ethiopia using EDHS 2016(n = 4083).** Source: Shape file is from CSA, 2013.

global Moran's I of 0.22 (p-value< 0.001). This showed that there was a significant clustering of age-appropriate vaccine practice in Ethiopia across the region. The spatial analysis also revealedthat the hotspot areas of age appropriate vaccination practice were clustered in Addis Ababa, Northern part of Tigray, south western part of Benishangul Gumuz, while the cold spot areas of age appropriate vaccination practice were found in Northern part of Somali, eastern part of Amhara, North eastern part of SNNPR, western part of Oromia and western part of Gambela. This may be the variation of health service availability and access to use, the variation of counseling service and level awareness about the appropriate schedule of vaccines across the region of Ethiopia. Additionally, Addis Ababa was predicted as more practice the age appropriate vaccination as compared to other regions. This may be associated with women live in

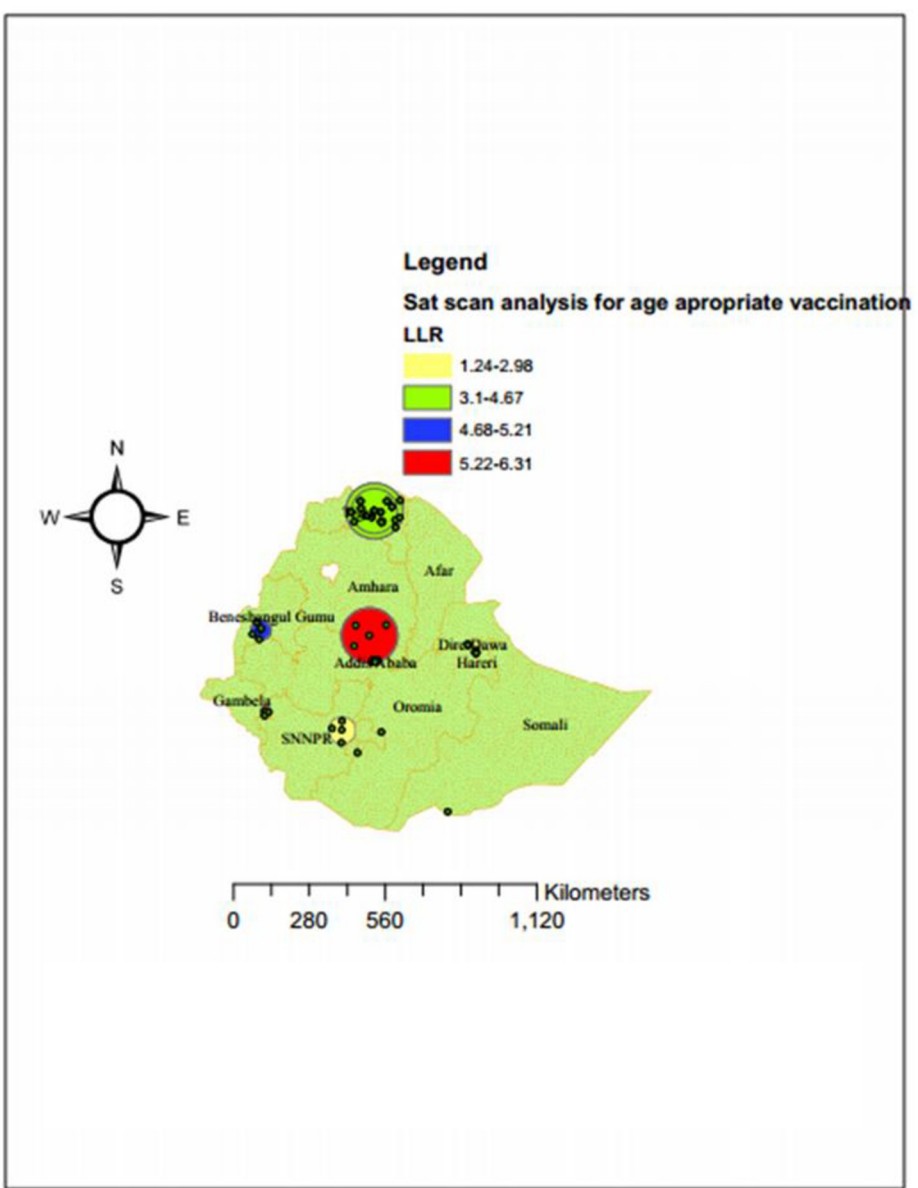

**Fig 7. Sat scan analysis of age appropriate vaccination practice in Ethiopia using EDHS 2016(n = 4083).** Source: Shape file is from CSA, 2013.

urban areas like in Addis Ababa are more informed about the appropriate schedule of vaccination as well as its advantage and easily access the vaccine timely than other parts of Ethiopia. Again in the SaT Scan analysis, the most likely clustered area of age appropriate vaccination practice was in Amhara region, which was 5.4 times higher than outside the window. This might be because of the difference in the awareness level of timely vaccinations inthe regions of Ethiopia.

This study also revealed that ANC utilization, receiving TT vaccine, wealth index, residence, and place of delivery were independently associated with age-appropriate vaccination practice. This study also revealed that the prevalence of age-appropriate vaccination practice was 33.8%

**Table 4. Significant clusters for age appropriate vaccination in Ethiopia using 2016 EDHS data (n = 4083).**

| Cluster type | Significant Enumeration Areas (clusters) detected | Coordinate/radius | Population | Cases | RR | LLR | p-value | P-value |
|---|---|---|---|---|---|---|---|---|
| Primary | 70, 349, 304, 88, 320, 621, 124, 161, 294, 65, 184, 335, 244, 569, 183, 150, 558, 137, 175, 364, 17, 36, 374, 494, 411, 280, 35, 563, 416 | (9.519402 N, 35.863468 E) / 131.73 km | 267 | 109 | 5.4 | 20.5 | 0.0000012 | <0.001 |
| Secondary | 392, 143, 136 | (12.876738 N, 39.326675 E) / 22.26 km | 18 | 14 | 3.8 | 11.8 | 0.0034 | <0.05 |
| Tertiary | 618, 266, 309, 435, 536, 370, 507, 592, 260, 104, 233, 69, 426, 603, 346, 315, 13, 567, 343, 105, 417, 284, 106, 265, 593 | (8.238420 N, 33.229507 E) / 147.30 km | 214 | 73 | 2.6 | 6.3 | 0.0028 | |
| Quarterly | 575, 579, 538, 355, 424, 430 | (13.508104 N, 38.958351 E) / 42.25 km | 72 | 28 | 4.8 | 4.22 | 0.0097 | |

[95% CI; 32.3, 35.3]. This finding is consistent with a study done in Debre Libanos North Shewa of the Oromia region (33.7%) [24]. and the study from Gondar city, north-west Ethiopia of 31.9% [25].

However, this finding is higher than a study done in the Menz Lalo district of Northeast Ethiopia of 6.2% [13] and 23.9% in the Toke Kutaye district, central Ethiopia [22]. This

**Table 5. Determinants of age appropriate vaccination practice among Ethiopian women using EDHS 2016 data (n = 4083).**

| Variables | Null model | Model I | Model II | Model III |
|---|---|---|---|---|
| **Residence** | | | | |
| Rural | | | 0.62 (0.5,0.76)* | 0.77(0.62,0.96)* |
| Urban | | | Reference | Reference |
| **Place of delivery** | | | | |
| Health facility | | | 1.84 (1.5,2.3)* | 1.2 (1.02,1.37)* |
| Home | | | Reference | Reference |
| **ANC utilization** | | | | |
| Yes | | 1.8 (1.5, 2.2)* | | 1.74 (1.45, 2.1)* |
| No | | Reference | | Reference |
| **# TT dose vaccine** | | | | |
| ≥ 2 | | 1.48 (1.22, 1.79)* | | 1.47 (1.22, 1.79)* |
| 1 | | 1.2 (0.85,1.6) | | 1.2 (0.85,1.58) |
| No vaccine take | | Reference | | Reference |
| **Wealth index** | | | | |
| Richest | | 2.1 (1.67, 2.8)* | | 1.65 (1.15, 2.37)* |
| Richer | | 1.79(1.36, 2.36)* | | 1.72 (1.32, 2.22)* |
| Middle | | 1.69(1.2, 2.4)* | | 1.65 (1.59, 2.35)* |
| Poorer | | 1.5 (1.13, 2.2)* | | 1.47 (1.11, 1.95)* |
| Poorest | | Reference | | Reference |
| **Own mobile phone** | | | | |
| Yes | | 1.2 (0.88, 1.55) | | 1.12 (0.85, 1.48) |
| No | | Reference | | Reference |
| **Care taker of child** | | | | |
| Other than mother | | 0.55 (0.24,1.21) | | 0.55 (0.25, 1.2) |
| Mother | | Reference | | Reference |

Note:

* statistically significant at p-value <0.05

discrepancy may be because of variations in the study period and the sample size. But the finding in the current study is lower than studies done in the pastoral community of Afar region, Ethiopia 43.7% [26], studies conducted in Uganda, 45.6% [27], China 43.72% to 59.25% [28], Tanzania 82.7% to 88.5% [29], Saudi Arabia 73% [30], Ghana 87.3% [31], Cameron 73.3% [32], Kenya 71–91% [33], Gambia 36.7% [34], and South Africa 58% to 88% [35]. This discrepancy may be associated with the variation of awareness about the schedule of age appropriate vaccine, variation of public awareness about timely vaccination and variation in cultural or socioeconomic status across the country.

The odds of the age-appropriate vaccine among those who have a history of ANC follow-up were more likely than among women without ANC follow-up. This is supported by studies done in Tanzania [32], Sinana District, Southeast Ethiopia [36], Ambo woreda, Central Ethiopia [37] and north Shewa Oromia Ethiopia [37]. This could be because of women who get ANC follow-up can get advice about the vaccination programs and its advantages than no ANC follow-up. Similarly, women who lived in the rural area decreased the odds of age-appropriate vaccination practice. This is supported by a study conducted in Ghana [8]. This may be related to women from a rural areas is vary in health service-seeking behavior, knowledge about vaccination programs, and inaccessibility of the service. The economic status of women or women with better economic status have high odds of age age-appropriate vaccination practice than poorest women [8]. This is supported by a study conducted in Ghana [31]. This could be because of a better socioeconomic status can be an important factor for accessing the health service easily or women who had a better income could have better health care-seeking behaviors about vaccines [38]. Women who gave birth at health facilities also have high odds of age age-appropriate vaccination practice than women who give birth at home. This finding is consistent with a study conducted in Debre Libanos district North Shewa Oromia regional state [24]. The possible reason may be associated with women who give birth at health facilities is the high possibility of getting adequate information about the appropriate schedule of the basic vaccines and health education than delivering at home [22]. Furthermore, women who receive two or more doses of the TT vaccine are more likely to vaccinate their child at the appropriate age than never receive the TT vaccine. This could be because women who receive the TT vaccine may provide a good opportunity to contact the health care providers to know about advantage of vaccines. Furthermore, those who receive more TT vaccine doses may have good health service-seeking behavior. In return, they apply this behavior to their child. National-based survey with a large amount of sample was the strength of the study. But unable to show the temporality relationship and missing the most important variable were the limitations of the study.

## Conclusion

The spatial variation of age appropriate vaccination was non-randomly distributed in Ethiopia across the region. Age-appropriate vaccination practice was low in Ethiopia compared to the WHO guideline. Wealth index, usual caretaker of the child, ANC utilization, history of TT vaccination, place of delivery, and residence were associated with age-appropriate vaccination practice. The finding has several implications for the healthcare system. So health facilities should better to work hard to increase the age-appropriate vaccination practice by expanding counseling and other health services. Providing ANC service, encouraging the mother to give birth in the health facility, giving health education about the advantage and the schedule of vaccines.

## Acknowledgments

The authors acknowledge DHS international for providing this data set.

## Author Contributions

**Conceptualization:** Muluken Chanie Agimas, Meron Asmamaw, Nebiyu Mekonen, Fantu Mamo, Daniel Alayu Shewaye.

**Data curation:** Muluken Chanie Agimas, Meron Asmamaw, Nebiyu Mekonen, Fantu Mamo, Daniel Alayu Shewaye.

**Formal analysis:** Muluken Chanie Agimas, Meron Asmamaw, Nebiyu Mekonen, Fantu Mamo, Daniel Alayu Shewaye.

**Funding acquisition:** Daniel Alayu Shewaye.

**Investigation:** Muluken Chanie Agimas, Fantu Mamo.

**Methodology:** Muluken Chanie Agimas, Fantu Mamo.

**Software:** Muluken Chanie Agimas.

**Supervision:** Muluken Chanie Agimas.

**Validation:** Muluken Chanie Agimas.

**Writing – original draft:** Muluken Chanie Agimas.

**Writing – review & editing:** Muluken Chanie Agimas.

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
