## [Decision Letter · Decision Letter 0]

22 Feb 2023

PONE-D-23-00874Spatial variation and determinants of age-appropriate vaccination practice in Ethiopia using EDHS 2016 data: A multilevel and spatial analysisPLOS ONE

Dear Dr. chanie,

Thank you for submitting your manuscript to PLOS ONE. After careful consideration, we feel that it has merit but does not fully meet PLOS ONE’s publication criteria as it currently stands. Therefore, we invite you to submit a revised version of the manuscript that addresses the points raised during the review process.

We look forward to receiving your revised manuscript.

Kind regards,

Gizachew Gobebo Mekebo, Assistant Professor

Academic Editor

PLOS ONE

Journal Requirements:

"no specific funding for this work"

6. We note that Figures 1, 4, 5, 6 and 7 in your submission contain map/satellite images which may be copyrighted. All PLOS content is published under the Creative Commons Attribution License (CC BY 4.0), which means that the manuscript, images, and Supporting Information files will be freely available online, and any third party is permitted to access, download, copy, distribute, and use these materials in any way, even commercially, with proper attribution. For these reasons, we cannot publish previously copyrighted maps or satellite images created using proprietary data, such as Google software (Google Maps, Street View, and Earth). For more information, see our copyright guidelines: http://journals.plos.org/plosone/s/licenses-and-copyright.

a. You may seek permission from the original copyright holder of Figures 1, 4, 5, 6 and 7 to publish the content specifically under the CC BY 4.0 license.  

Additional Editor Comments:

Your manuscript has been reviewed by two reviewers. Based on the review reports, and my own assessment as Editor, I am pleased to inform you that it has a potential to be accepted for publication in PLOS ONE, once you have significantly carried out revisions suggested by the reviewers. Therefore, we invite you to significantly revise the manuscript.

Reviewers' comments:

Reviewer's Responses to Questions

**Comments to the Author**

1. Is the manuscript technically sound, and do the data support the conclusions?

Reviewer #1: Yes

Reviewer #2: Yes

 2. Has the statistical analysis been performed appropriately and rigorously?

Reviewer #1: Yes

Reviewer #2: Yes

3. Have the authors made all data underlying the findings in their manuscript fully available?

Reviewer #1: Yes

Reviewer #2: Yes

 4. Is the manuscript presented in an intelligible fashion and written in standard English?

Reviewer #1: No

Reviewer #2: Yes

 5. Review Comments to the Author

Reviewer #1: The authors tried to assess the spatial variation and determinants of age-appropriate vaccination practice in Ethiopia using EDHS 2016 data. Then they concluded as the spatial distribution of age-appropriate vaccination practice in Ethiopia was non-randomly distributed across the regions. Age-appropriate vaccination practice was low in Ethiopia. Wealth index, usual caretaker of the child, ANC utilization, history of TT vaccination, place of delivery, and residence were associated with age-appropriate vaccination practice. The idea of good.

• The manuscript needs to take an editorial revision to tackle minor grammar issues.

• I suggest that the authors consider more updated references to the introduction subsection.

• To clearly indicate and just the existence of the spatial variation, it is also highly important if some spatial tests are taken into account. With these comments, I recommend that the article be published.

• I don’t see the effects of age when dealing the spatial issues. What is the relevance of considering age as one of the explanatory variables in dealing with spatial analysis?

• I also suggest the authors to highlight and explain the issue of spatial dependency in the methodology subsection.

Regards

Habte Tadesse, PhD

Department of statistics, Addis Ababa University, Ethiopia

Reviewer #2: Comments to author(s)

Manuscript Number: PONE-D-23-00874

Title: Spatial variation and determinants of age-appropriate vaccination practice in Ethiopia using EDHS 2016 data: A multilevel and spatial analysis

I appreciate the authors for their effort to conduct this study. The manuscript reports findings based on the objective of the study. The overall organization of the manuscript is also good. But, there are some concerns that need corrections. The comments are listed below, review and amend them.

Introduction

Some of your references cited in the introduction part has no relation to your document.

Citation #1 talks about “The impact of the vaccination program for hemorrhagic fever with renal syndrome in Hu County, China”, remove this citation and replace it with appropriate citation.

Citation #2 briefs about influenza vaccine not about period of vaccine administration. Revise and replace with the appropriate citations.

Citation #3 is not the finding of the research you cited. Rather, you used from cross reference. So, replace it with appropriate source,

Page 3, line 58-59; revise the sentence. Next to citation 6, please add the program of vaccination based on WHO recommendation.

Page 4, line 71; write full words on their first appearance, VPDs.

Page 4, line 76; the citation used for Toke kutaye reference number 18 is not correct. The reference listed on number 18 is Xiao X, Wu Z-C, Chou K-C , 2011.

Methods

Page 4, line 84-85; the population of Ethiopia is 114,963,588, include your reference

Page 4, line 89-90; what is the need of this statement “the Ethiopian demographic health survey of 2016 revealed that among ten children aged 12-23, four children (39%) received all the basic vaccines (19).:”?

Page 6, line 119; table 1, there is no table clearly showing table 1 since table cations are 1 for all.

Results

Page 7, line 144, line 149, and page 9 line 200; you cited table 2,3, and 4, but not appropriately used the captions.

Page 8, line 162; figure 4 is not available in lists of tables or has no caption.

Pages 7- 8, line 158-162; why you included about anemia? Remove it.

Discussion

At the end of your discussion add strength and limitation of your study.

Conclusion

Actually, you have no variable that shows lack of transportation fees. So, you could not give the following recommendation since it is not a part of your findings. So remove it. “provide outreach vaccination for those are difficulty accessing the vaccine because of lack of transport fees could bring a significant contribution to improving age appropriate vaccination practice by health care workers.”

Page 12, line 264-266; move the strength and limitation to discussion part.

References

References 1-3 are not related with your citation, so change and update them.

Reference number 19 is not fully written. Revise it.

Tables

There are 4 tables listed in the manuscript. But the caption of each table is not correct. All tables are labeled as “table 1”, revise and correct them.

Sociodemographic table, under variable region, change Somalia to Somali.

Figures

Out of the 7 figures listed in in this manuscript, one figure has no caption. Revise it.

 6. PLOS authors have the option to publish the peer review history of their article (what does this mean?). If published, this will include your full peer review and any attached files.

Reviewer #1: **Yes: **ok

Reviewer #2: No

---

## [Author Response · Author response to Decision Letter 0]

24 Oct 2023

Spatial pattern and associated factors of timely vaccination in Ethiopia using EDHS 2016 data: a spatial and multilevel analysis. 

Muluken Chanie Agimas*1, Meron Asmamaw1, Nebiyu Mekonen1, Fantu Mamo1, Daniel Alayu Shewaye1.

1. College of medicine and health science, institute of public health, department of epidemiology and biostatistics, university of Gondar, Gondar, Ethiopia. 

Authors address: 

Muluken Chanie: mulukensrc12@gmail.com

Meron Asimamaw: merryalem101@gmail.com

Nebiyu Mekonen: nebiyumek12@gmail.com

Fantu mamo: fantuma3@gmail.com

Daniel Alayu: danielalayu14@gmail.com

Corresponding Author 

Muluken Chanie: mulukensrc12@gmail.com

To: PLOS ONE <em@editorialmanager.com>

 Manuscript Number: PONE-D-23-00874

Cover Letter for reviewer’s comments and authors responses 

We would like to thank both the Editorial board members and reviewers for kind comments and suggestions. Accordingly, we have tried to correct and respond to those comments point by point, page by page, and line by line. Therefore, we prepared this author’s response, the revised manuscript, the separate revised figures and tables, and the tracked changes of the manuscript. 

Sincerely, with best regards! 

Muluken Chanie Agimas 

ID: https://orcid.org/0000-0003-0808-124X

 Comments from review and editorial board Authors response 

1. Some of your references cited in the introduction part has no relation to your document. 

a. Citation #1 talks about “The impact of the vaccination program for hemorrhagic fever with renal syndrome in Hu County, China”, remove this citation and replace it with appropriate citation. 

b. Citation #2 briefs about influenza vaccine not about period of vaccine administration. Revise and replace with the appropriate citations.

c. Citation #3 is not the finding of the research you cited. Rather, you used from cross reference. So, replace it with appropriate source, 

Dear reviewer it is a very interesting comment and we accepted, so that we removed those references which are inappropriately cited and substituted with the appropriate references so we revised our manuscript accordingly.

2. Page 3, line 58-59; revise the sentence. Next to citation 6, please add the program of vaccination based on WHO recommendation.

 Great comments: totally accepted and has been revised as follows:

the world health organization (WHO) recommends that the vaccine should take within 1-year schedule to maximize the protective effect of the vaccine

3. Page 4, line 71; write full words on their first appearance, VPDs Great. Corrected as follows on page 4 line-71

Like other countries, in Ethiopia, age-appropriate vaccination is the key strategic activity to reduce easily preventable childhood mortality from measles, pneumonia, diarrheal diseases, and other vaccine preventable diseases (17). 

4. Page 4, line 76; the citation used for Toke kutaye reference number 18 is not correct. The reference listed on number 18 is Xiao X, Wu Z-C, Chou K-C , 2011.

 Thank you dear reviewer you observe wisely we saw the reference and really needs amendment. So the comment is corrected by the following reference 

 19. Dirirsa K, Makuria M, Mulu E, Deriba BS. Assessment of vaccination timeliness and associated factors among children in Toke Kutaye district, central Ethiopia: A Mixed study. Plos one. 2022;17(1):e0262320.

 Method 

5. Page 4, line 84-85; the population of Ethiopia is 114,963,588, include your reference 

 Well as long as we use another’s report we have to acknowledge the source so we admit the comment and revision has made on references and the population size of Ethiopia (latest data)

6. Page 4, line 89-90; what is the need of this statement “the Ethiopian demographic health survey of 2016 revealed that among ten children aged 12-23, four children (39%) received all the basic vaccines (19).:”? 

Dear reviewer it is a very interesting comment and we accepted, so that we removed those phrasing which are irrelevant to the study 

7. Page 6, line 119; table 1, there is no table clearly showing table 1 since table captions are 1 for all. 

 Great comment and correction has made by the following ways:

Intra-class coefficient (ICC) was used for the measurement of variation across regions with value of 18.7%. Analysis was undergone in four models; null model (model of without independent variables), model-I (individual level predictors), model-II (group level predictors), model-III (both model-I and model-II). For all models ICC, a proportional change in variance (PCV), log-likelihood test, and Akaike information criterion (AIC) model were calculated (Table-1).

 Results 

8. Page 7, line 144, line 149, and page 9 line 200; you cited table 2, 3, and 4, but not appropriately used the captions. 

9. Page 8, line 162; figure 4 is not available in lists of tables or has no caption. Great view and revision also made on the table captions. 

Great view and revision also made on the figure-4 (figure-3 in the revised figure) caption. And the list will be incorporated when submitted 

10. Pages 7- 8, line 158-162; why you included about anemia? Remove it. 

It is an editorial problem and the revision has made and replaced with age appropriate vaccination practice. 

11. At the end of your discussion add strength and limitation of your study. 

 Dear reviewers, if the journal recommends the limitation and the strength of the study as a part of the discussion, we will do it. 

 Conclusion 

12. Actually, you have no variable that shows lack of transportation fees. So, you could not give the following recommendation since it is not a part of your findings. So remove it. “Provide outreach vaccination for those are difficulty accessing the vaccine because of lack of transport fees could bring a significant contribution to improving age appropriate vaccination practice by health care workers.”

 Okay It is obvious that the recommendations should be based on the findings. So revision has made accordingly 

13. Page 12, line 264-266; move the strength and limitation to discussion part. 

 Dear reviewers, as we have mention earlier in comment 12:

 If the journal recommends the limitation and the strength of the study as a part of the discussion, we will do it. 

 References 

14. References 1-3 are not related with your citation, so change and update them.

 Great. The necessary revision has made because the reference was unrelated 

15. Reference number 19 is not fully written. Revise it. Dear reviewer as we have stated earlier it is a very interesting comment and we accepted it. So that we removed those references which are inappropriately cited and substituted with the appropriate references so we revised our manuscript accordingly.

 Tables 

16. There are 4 tables listed in the manuscript. But the caption of each table is not correct. All tables are labeled as “table 1”, revise and correct them. Okay accepted. The caption and the labels of each table has been corrected After the correction, the revised tables and figures will be attached accordingly 

17. Sociodemographic table, under variable region, change Somalia to Somali. 

 Thank you for your detailed comments, all the comments that you have raised are very important and we just accept all the comments and we have taken a lesson from the comments. Similarly The necessary amendment has made on comment_17. Which is Somalia is changed to Somali in the revised manuscript. 

 Figures 

18. Out of the 7 figures listed in in this manuscript, one figure has no caption. Revise it. 

 Definitely. Figure-4 (figure-3 in the revised figure) has no caption. So amendment/revision has made. In the revision the most important comments of the captions are incorporated. 

Other comments from the reviewer 

The manuscript needs to take an editorial revision to tackle minor grammar issues. Dear reviewer the grammar error has been reviewed. 

To clearly indicate and just the existence of the spatial variation, it is also highly important if some spatial tests are taken into account. With these comments, I recommend that the article be published. Dear reviewer, you are right. Definitely, if spatial tests were added it was better. But in this study, our objective was to show the geographical distribution/variation of timely vaccination practice. Therefore we have reported the finding based on the stated objectives.

I don’t see the effects of age when dealing the spatial issues. What is the relevance of considering age as one of the explanatory variables in dealing with spatial analysis? Dear reviewer in my research an attempt has been made to show the distribution of timely vaccination/age-appropriate vaccination across the region of Ethiopia. So age in this case shows whether the vaccine was taken at the appropriate age or not (Yes, No). For example, according to WHO, BCG, the polio-0 vaccine should take at birth, polio-1, penta-1 pcv-1 should be taken at one month or after 4 weeks and so on. Therefore in this study we show the spatial distribution of this timely/age appropriate vaccination. Not mean that the spatial variation of the child age.

I also suggest the authors to highlight and explain the issue of spatial dependency in the methodology subsection.

 Great comment. We miss it and now we revise it accordingly.

Comments from editors 

In your Data Availability statement, you have not specified where the minimal data set underlying the results described in your manuscript can be found. PLOS defines a study's minimal data set as the underlying data used to reach the conclusions drawn in the manuscript and any additional data required to replicate the reported study findings in their been entirety. The correction has been made based on the comment. And the supporting information has been submitted by the file name “supporting information”

We note that you have stated that you will provide repository information for your data at acceptance. Should your manuscript be accepted for publication, we will hold it until you provide the relevant accession numbers or DOIs necessary to access your data. If you wish to make changes to your Data Availability statement, please describe these changes in your cover letter and we will update your Data Availability statement to reflect the information you provide. My ID= https://orcid.org/0000-0003-0808-124X

We note that Figures 1, 4, 5, 6 and 7 in your submission contain map/satellite images which may be copyrighted. All PLOS content is published under the Creative Commons Attribution License (CC BY 4.0), which means that the manuscript, images, and Supporting Information files will be freely available online, and any third party is permitted to access, download, copy, distribute, and use these materials in any way, even commercially, with proper attribution. For these reasons, we cannot publish previously copyrighted maps or satellite images created using proprietary data, such as Google software (Google Maps, Street View, and Earth). For more information, see our copyright guidelines Dear editor deadly sure the image never copy from other sources rather the image was from ARC GIS output. But the shape file was from CSA 2013. Therefore the source of the shape file included in the revised shape/figure 

But figure one was obtained from other sources and I cannot obtain permission from the original copyright holder to publish these figures under the CC BY 4.0 license. So I remove the figure -1

 Dear editor this comment is not only your comment but also the reviewers’. Therefore I accept the comment and an amendment has been made.

---

## [Decision Letter · Decision Letter 1]

7 Dec 2023

Spatial pattern and associated factors of timely vaccination in Ethiopia using EDHS-2016 data: A multilevel and spatial analysis

PONE-D-23-00874R1

Dear Dr. chanie,

We’re pleased to inform you that your manuscript has been judged scientifically suitable for publication and will be formally accepted for publication once it meets all outstanding technical requirements.

Kind regards,

Gizachew Gobebo Mekebo

Academic Editor

PLOS ONE

Additional Editor Comments (optional):

Reviewers' comments:

Reviewer's Responses to Questions

**Comments to the Author**

1. If the authors have adequately addressed your comments raised in a previous round of review and you feel that this manuscript is now acceptable for publication, you may indicate that here to bypass the “Comments to the Author” section, enter your conflict of interest statement in the “Confidential to Editor” section, and submit your "Accept" recommendation.

Reviewer #2: (No Response)

2. Is the manuscript technically sound, and do the data support the conclusions?

Reviewer #2: Yes

3. Has the statistical analysis been performed appropriately and rigorously? 

Reviewer #2: Yes

4. Have the authors made all data underlying the findings in their manuscript fully available?

Reviewer #2: Yes

5. Is the manuscript presented in an intelligible fashion and written in standard English?

Reviewer #2: Yes

6. Review Comments to the Author

Reviewer #2: (No Response)

7. PLOS authors have the option to publish the peer review history of their article (what does this mean?). If published, this will include your full peer review and any attached files.

Reviewer #2: No

---

## [Editor Report · Acceptance letter]

25 Jan 2024

PONE-D-23-00874R1 

PLOS ONE

Dear Dr. Agimas, 

I'm pleased to inform you that your manuscript has been deemed suitable for publication in PLOS ONE. Congratulations! Your manuscript is now being handed over to our production team.

Kind regards, 

on behalf of

Assistant Professor Gizachew Gobebo Mekebo 

Academic Editor

PLOS ONE